# Distributed Fine-tuning of Language Models on Private Data

**Vadim Popov, Mikhail Kudinov, Irina Piontkovskaya, Petr Vytovtov & Alex Nevidomsky**
Samsung R&D Institute Russia
Moscow, Russia
`v.popov@samsung.com,m.kudinov@samsung.com,`
`p.irina@samsung.com,p.vytovtov@partner.samsung.com,`
`a.nevidomsky@samsung.com`

## Abstract

One of the big challenges in machine learning applications is that training data can be different from the real-world data faced by the algorithm. In language modeling, users' language (e.g. in private messaging) could change in a year and be completely different from what we observe in publicly available data. At the same time, public data can be used for obtaining general knowledge (i.e. general model of English). We study approaches to distributed fine-tuning of a general model on user private data with the additional requirements of maintaining the quality on the general data and minimization of communication costs. We propose a novel technique that significantly improves prediction quality on users' language compared to a general model and outperforms gradient compression methods in terms of communication efficiency. The proposed procedure is fast and leads to an almost 70% perplexity reduction and 8.7 percentage point improvement in keystroke saving rate on informal English texts. Finally, we propose an experimental framework for evaluating differential privacy of distributed training of language models and show that our approach has good privacy guarantees.

## 1 Introduction

Two common problems arising after deployment of a machine learning model on user devices are discrepancy between training data and actual data stored on user devices, and the need of regular model updates. In the case of language modeling, it corresponds to the difference between language and style of the training corpus mined in the Internet and messages of the user, which account for most of the text generated on the device. Even if the training corpus includes a substantial part of informal texts (tweets, forum threads, etc.), real user data can be very different. This is a challenge for word prediction algorithms in software keyboard applications. The most general approach to improvement of customer experience in typing is integrating a separate user language model trained on device in an on-line fashion. In the simplest case it is a smoothed n-gram (e.g. Kneser-Ney n-gram model (Goodman (2001))).

In Yoon et al. (2017) continuously learned personalized language model based on LSTM was proposed but as far as each user generates only a small portion of textual data, such data by itself cannot be used for updates of the general model. Thus, for a model update, a collection of potentially sensitive data from many users is needed. As shown in McMahan et al. (2016), collecting data for training may be avoided. We propose a similar approach for distributed fine-tuning of language models on private data. In this sense our method can be considered as "federated fine-tuning" but we prefer to take more traditional term. In this setting we start with a language model trained on a large text corpus representing the general language. This model $G$ will be updated continuously on user devices but with an additional requirement that the model must not go too far from the general language model, i.e. we don't overfit on user data.

We pursue two goals: 1) to develop an algorithm of distributed fine-tuning that is fast, communication efficient and doesn't need collecting sensitive user data; and 2) to prevent the language model from forgetting "general English". Besides, we provide analysis of possibility of privacy violation

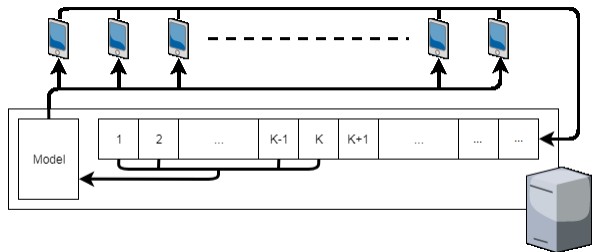

Figure 1: Overview of the approach. The current model is updated on devices and updates $\bar{G}_t^i$ from users are stored in a queue. Every $K$ elements $\bar{G}_t^i$ of the queue are used for one round of averaging. After each round the server model $G_{t+1}$ is sent to the next $K$ elements.

in our model. (Hitaj et al. (2017)) demonstrated an attack on distributed training algorithm leading to information leakage. This means that privacy analysis in necessary for such algorithms.

Our main contributions are: 1) we propose an efficient procedure of distributed fine-tuning of language models immune to the problem of catastrophic forgetting (French (1999)), 2) we provide experimental evaluation of on-device training time, communication costs and convergence rates of the general language model in realistic conditions, 3) we compare two most popular strategies of improving communication efficiency in the context of distributed learning, and 4) we propose an experimental framework for evaluation of differential privacy of distributed training of language models, and using this framework, we evaluate privacy guarantees of our approach.

In our research we are focused on improvement of keystroke saving rate (see section 2.4) because this metric reflects customer typing experience more directly than perplexity or BLEU. We use LSTM architecture for our language model as described in Zaremba et al. (2014) and evaluate on-device training time for this architecture. We show that the on-device training time is reasonably small, thus demonstrating the feasibility of the whole approach.

## 2 DISTRIBUTED FINE-TUNING OF LANGUAGE MODELS

As usual, our task is to predict the next word $w_N$ given a sequence of words $w_1 \ldots w_{N-1}$. If the prediction algorithm of a software keyboard application is based on a language model with low perplexity on the test data, the application provides a reasonably sorted list of input candidates. Of course, the test data should be drawn from the same distribution as the user data. In our case we also want to have only one, continuously improving model on a user device. As far as the user can always switch to the general English, we have to prevent the model from overfitting on the texts written on the device, or *catastrophic forgetting* (McCloskey & Cohen (1989); Goodfellow et al. (2014); Kirkpatrick et al. (2016)).

Our approach can be summarized as follows (Figure 1): 0) At the first stage we have an initial language model $G_0$ (at every step t it will be updated to $G_t$) trained on a large corpus of standard English; 1) As soon as a user inputs sufficient volume of text, the latest version of $G_t$ is sent from the server to provide updates, and fine-tuning starts on the device leading to the model $\bar{G}_t^i$; 2) When the training is finished the model $\bar{G}_t^i$ is sent back to the server; 3) Every time the updated models $\bar{G}_t^i$ are received from $K$ different users, a round of model update is run resulting in the model $G_{t+1}$

### 2.1 LEARNING WITHOUT FORGETTING

In its original formulation (Li & Hoiem (2016)), the problem of learning without forgetting (LwF) consists in re-training of existing model $\Theta$ on new data such that its performance on the old data does not degrade.

More formally, suppose we have a classifier with a set of parameters $\Theta$ trained and tested on a dataset $\mathbf{D} = \{\mathbf{Tr}, \mathbf{Ts}\}$ where $\mathbf{Tr}$ and $\mathbf{Ts}$ are train and test sets accordingly. Let $\mathbf{D}^* = \{\mathbf{Tr}^*, \mathbf{Ts}^*\}$ be some new dataset. Our goal is to update the parameters $\Theta$ with dataset $\mathbf{D}' = \{\mathbf{Tr}^*, \mathbf{Ts} \cup \mathbf{Ts}^*\}$

i.e. we have to provide the best performance on old and new types of data having only training data of the new type.

In contrast, joint training (Caruana (1997)) assumes a model update with access to the both datasets: $\mathbf{D}' = \{\mathbf{Tr} \cup \mathbf{Tr}^*, \mathbf{Ts} \cup \mathbf{Ts}^*\}$.

As we want to avoid sending user data to the server, classical joint training is impossible. On the other hand, LwF seems promising. In this case we send the user a current instance of the general language model $G_t$ with weights $\theta_g$ and fine-tune it producing the model $\theta_u$, while $\theta_g$ is used for generating predictions for regularization. The resulting loss at step $t$ and true word $\mathbf{w}_t$ can be calculated as follows:

$$l_t(\theta_u) = - \sum_{w \in W} y_{t,w}^* \log p(w|\theta_u),$$ (1)

where

$$y_{t,w}^* = \lambda \mathbf{1}\{\mathbf{w}_t = w\} + (1 - \lambda)p(w|\theta_g)$$ (2)

A similar approach is taken in Shin et al. (2016) where predictions of a basic model (in this case $\theta_g$) are taken as *soft labels*.

## 2.2 TRAINING WITH REHEARSAL

Minimizing loss in (1)–(2) is equivalent to minimizing Kullback-Leibler divergence $\mathcal{L}(\theta_u) = KL(\mathbb{P}_{gr}\|\mathbb{P}_u)$ with respect to parameters $\theta_u$ of $\mathbb{P}_u$ where density of $\mathbb{P}_{gr}$ is given by:

$$P(x) = \lambda P_{Tr^*}(x) + (1 - \lambda)P(x|\theta_g)$$ (3)

In (3) $P_{Tr^*}(x)$ stands for the real distribution on a user device and $P(x|\theta_g)$ is a probability given by the model of "general English" $\theta_g$. It suggests that instead of optimizing $\mathcal{L}(\theta_u)$ we can simply add data from $\mathbf{Tr}$ to $\mathbf{Tr}^*$ to obtain the $(1 - \lambda)$ portion. This approach, called *random rehearsal*, was presented in Robins (1995).

In practice in the case of fine-tuning with rehearsal a portion of the general English training corpus (standard English corpus) must be sent to the user device. Volume of typical user data generated on device is of the order of tens of kilobytes per month, and the size of the training data sent to the device will be of the same order. Overall, random rehearsal is more efficient, because there is no need to calculate soft labels.

## 2.3 SERVER-SIDE MODEL UPDATE

The server-side part of the solution must aggregate models $\bar{G}_t^i$ from many users and use them to update the general model $G_t$. We took simple model averaging as a baseline solution and transfer learning (Bengio (2011); Tang et al. (2016)) as an alternative approach.

In the case of transfer learning we optimized cross-entropy function (1), with $y_i^*$ given by an average prediction from $N$ aggregated models $\theta_u^k$:

$$y_i^* = \frac{1}{N} \sum_{k=1}^{N} p(w_i|\theta_u^k)$$ (4)

Just as in the case of on-device training, transfer learning-based approach is rather inefficient in terms of time and memory because predictions from all models are needed.

## 2.4 KEYSTROKE SAVING RATE

Keystroke saving rate (KSS) (McKenzie & Soukoreff (2002)) is defined as a relative decrease in the number of characters the user has to type, given suggestions from the software keyboard:

$$KSS = \frac{N_{total} - N_{typed}}{N_{total}} \times 100\%,$$ (5)

Table 1: Random rehearsal vs learning without forgetting. For LwF mode $\lambda$ is a coefficient of the ground truth probability distribution in the loss function (1)-(2). For random rehearsal mode $\lambda$ is a portion of user training data in on-device training.

| Method | Standard English dataset (Wikipedia) | | User dataset (Twitter) | | Av. PPL |
|---|---|---|---|---|---|
| | PPL | KSS, % | PPL | KSS, % | |
| Initial server model | 100.1 | 67.9 | 336.0 | 49.7 | 192.6 |
| Random rehearsal, $\lambda = 1/4$ | 121.3 | 66.3 | 127.9 | 56.9 | 124.8 |
| Random rehearsal,$\lambda = 1/2$ | 131.1 | 65.9 | 109.7 | 58.3 | **119.1** |
| Random rehearsal,$\lambda = 3/4$ | 149.0 | 64.8 | 99.7 | 59.0 | 119.9 |
| Learning without forgetting, $\lambda = 1/4$ | 128.4 | 66.0 | 162.8 | 54.9 | 146.0 |
| Learning without forgetting, $\lambda = 1/2$ | 147.0 | 64.9 | 121.7 | 57.5 | 132.7 |
| Learning without forgetting, $\lambda = 3/4$ | 186.5 | 63.1 | 101.1 | 59.2 | 133.9 |
| On-device re-training, $\lambda = 1$ | 265.1 | 60.2 | 93.4 | 59.7 | 150.8 |

where $N_{total}$ is the total number of non-space characters in the typed text and $N_{typed}$ is the number of characters user still had to type until the correct suggestion was presented. In our experiments we used top-3 suggestion lists.

From the definition above one can see that KSS is better for customer experience assessment compared to perplexity. Besides, perplexity measure underestimates out-of-vocabulary (OOV) words. In the presence of OOV words perplexity is ill-defined, so all OOV words must be removed from the test set. It makes a direct comparison of models with different vocabularies impossible, which is impractical. Finally, our experiments have demonstrated that a small decrease in perplexity may not correspond to KSS improvement and doesn't lead to any practical result. Nevertheless, our method demonstrates considerable perplexity reduction as well.

## 2.5 MODEL FINE-TUNING EXPERIMENTS

The goal of our experiments was to find the most efficient pipeline to distributed fine-tuning of language models. We compared several approaches for client-side and server-side model updates. In accordance with the problem statement we assumed a substantial difference between the real-life user corpus and the standard English corpus used for initial training, so we took Twitter and Wikipedia corpora for the user and standard English corpora correspondingly.

The standard English train dataset contained approximately 30M tokens. The hyperparameters of the model were initially tuned on the Standard English validation set of 3.8M tokens. The user train dataset contained approximately 1.7M tokens. Updated models were tested on subsets of the Twitter and Wikipedia corpora containing 200k and 170k tokens correspondingly. Comparison between the random rehearsal and LwF training methods were carried out on a single node.

For our experiments we used LSTM architecture from Zaremba et al. (2014) with 2x650 LSTM layers, a vocabulary size of 30k, dropout $0.5$, minibatch size 20, BPTT steps 35. The initial general English model was trained in 39 epochs.

We report KSS and perplexity on both the standard English test set and the user data test sets. In the case of the standard English test set KSS was calculated on a subset of 200 sentences (3600 tokens). The initial general English model had a perplexity of $100.1$ and $67.9\%$ KSS rate on the Standard English test and perplexity $336.0$ and $49.7\%$ KSS rate on the user data test set. So, the model experienced a considerable $18.2\%$ drop in performance on the user data test set.

Table 1 summarizes our experiments with on-device model update algorithms. We see that the performance gap between the standard English and the user test sets can be considerably reduced at the cost of performance degradation on the first dataset. The best average perplexity is reached with the random rehearsal method and $\lambda = 0.5$. We believe that the reason of the comparably inferior performance of the LwF method can be explained by the fact that soft labels used by LwF give a poor approximation of the true word distribution of general English so adding a small portion of true data gives better results in terms of knowledge preservation.

Table 2: Averaging vs transfer learning for server-side model update.

| Method | Standard English dataset (Wikipedia) | | User dataset (Twitter) | | Av. PPL |
|---|---|---|---|---|---|
| | PPL | KSS, % | PPL | KSS, % | |
| Initial server model | 100.1 | 67.9 | 336.0 | 49.7 | 192.6 |
| TL on generated data (1-cycle) | 109.2 | 67.2 | 259.7 | 50.8 | 174.4 |
| TL on generated data (5-cycles) | 112.3 | 67.0 | 246.0 | 51.2 | 171.6 |
| TL on real data | 108.7 | 67.2 | 261.2 | 50.7 | 174.6 |
| Model averaging (1 round) | 102.8 | 67.7 | 233.8 | 51.9 | **160.3** |
| Model averaging (300 rounds) | 105.5 | 67.3 | 109.3 | 58.4 | **107.5** |

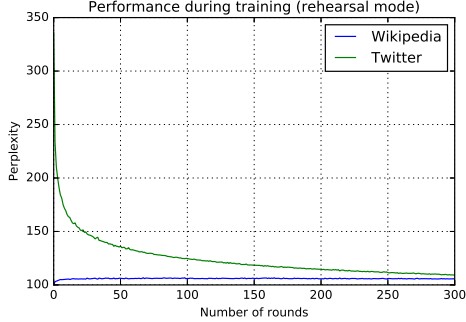 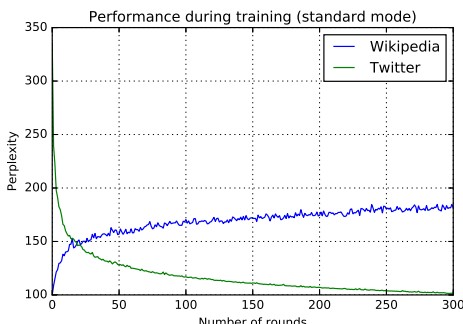

Figure 2: Training curves for the general model on the standard English (Wikipedia) and the user data (Twitter) corpora with random rehearsal (left) and without random rehearsal (right).

To compare model averaging and transfer learning for a server-side model update, we carried out a small experiment with 10 nodes and 1 iteration of the server-side update. Each model was trained on a mobile phone with a quad-core mobile CPU with a clock frequency 2.31 GHz. We used a minibatch size 10, number of BPTT steps 20, learning rate 0.75 and 1 epoch. Training took approximately 140 seconds on 20 kilobytes of text (user-generated and rehearsal data). Note that we used mobile CPU only, so computation time may be reduced by using mobile GPU. Due to absence of the frameworks that make backpropagation on a device possible we had to implement our own training on the phone. After training the updated user models were used for general model update on the server.

For the server-side model update algorithm we also tried the approach proposed in Shin et al. (2016). In this case the new model is trained on the texts generated by its previous round of update. We tested both 1 generation per epoch and a single time generation before the first epoch. We carried out at most 6 epochs so we had 1 and 5 cycles of text generation correspondingly.

Results of the experiment are summarized in Table 2. We saw no significant differences between transfer learning on real and generated data. The difference between transfer learning and averaging is more sound but still not large. At the same time model averaging is much more computationally efficient, as long as transfer learning requires calculation of labels from each of the teacher models. After 300 rounds of model updates with 3000 nodes (10 nodes per round) we ended up with an 8.7 absolute gain in KSS on the user data test with only a 0.6 absolute KSS drop on the standard English data test.

Figure 2 shows that the model starts to perform reasonably well after 100 rounds of updates. It also shows the importance of rehearsal for preventing catastrophic forgetting.

## 2.6 COMMUNICATION COSTS

There are several strategies that help to make distributed learning communication efficient. The most successful ones can be divided into two classes: 1) strategies that increase computation on nodes thus sending data to the server less frequently (McMahan et al. (2016)), and 2) strategies that

Table 3: Uploaded data analysis

| Number of parameters | Size of the model | Nodes per round | Uploaded data per round |
| --- | --- | --- | --- |
| $4.57 \cdot 10^7$ | 174.43Mb | 10 | 1.70Gb |

Table 4: Communication costs comparison

| Communication efficiency improving scheme | Perplexity | Uploaded data | Number of uploads |
| --- | --- | --- | --- |
| Several epochs of on-device training | 71.06 | 21.29Gb | 112 |
| DGC (Lin et al. (2017)) | 72.24 | 21.85Gb | $5.3 \cdot 10^4$ |

transmit only some part of data from devices to the server in a single round of averaging Lin et al. (2017); Konečný et al. (2016). One of the most impressive results was reached by the Deep Gradient Compression (Lin et al. (2017)). It belongs to the second class – its key idea is to send only the most important weight updates obtained during on-device training while accumulating the remaining ones in order to send them when the sum becomes large enough.

It was shown that Deep Gradient Compression method (DGC) allows to send a very small part of weight updates (0.1%) in a single round of averaging without loss in the quality of the model. For language modeling task, the gradient compression ratio of 462x was obtained using gradient accumulation strategy for small updates. However, DGC supposes that in each round of averaging only one user's model update is made for every node while methods from the first class increase computation on nodes to several epochs before model averaging. In our experiments (2.5) we chose to train models on devices for one epoch rather than using DGC-style strategy. As shown in Table 3, this results in a total amount of 1.7Gb of data transmitted from nodes to the server in a single round (this amount certainly depends linearly on the size of the model). We used a classical 2-layer LSTM model from Zaremba et al. (2014) but there are models that perform similarly or better but have less parameters (e.g. Inan et al. (2016), Press & Wolf (2017)), so in practice we can improve the results shown in Table 3.

To prove competitiveness of our approach, we made the experiment (see Table 4) in the settings presented in Lin et al. (2017). We compared two strategies for improving communication efficiency: increasing computation on nodes and DGC. The models were trained on a popular language modeling benchmark PTB. The neural network architecture (2-layer LSTM with 1500 units, tied input and output embeddings and variational dropout with probability 0.65) as well as the results for DGC were taken from Lin et al. (2017). As for the first strategy, we trained the model for 28 rounds. During the first round, a randomly initialized model was trained on the first node, then sent to the second node, trained there, and so on. When training on the last (fourth) node was finished, the updated model was sent to all four nodes and the second round started. The remaining 27 rounds were standard rounds of model averaging. We had to make the first round so specific because we needed to simulate some kind of "pretraining" (which the task itself didn't suggest) in order to make model averaging perform well. Since we had only one training corpus, no rehearsal was applied during training on nodes. The number of training epochs on a node and learning rate decreased from 10-20 and 1.0 correspondingly in the first rounds to 1-3 and 0.27 in the last ones. We used minibatch size 20 and 35 BPTT steps.

The first strategy achieved better perplexity with the same amount of data sent from nodes to the server compared to DGC. The important thing is that the number of communications for it was 112 which is much less than 53k for DGC. Since communication efficiency involves not only the data that is transmitted from devices to the server but also the time that is necessary to set up connections, we can conclude that increasing computation on nodes perfroms better in terms of communication efficiency than gradient compression methods. This is why we chose the first strategy in our approach. Moreover, in our scheme the data on a device is used only once and can be deleted after the on-device training whereas in DGC and many other distributed learning schemes the data on each device is used many times (once per epoch).

Certainly, the two classes of strategies for improving communication efficiency are not mutually exclusive – we can apply DGC or, for example, methods that are described in Konečný et al. (2016) to further reduce communication costs but this is out of the scope of the present paper.

# 3  PRIVACY ANALYSIS

## 3.1  METHODOLOGY

Our analysis is based on the experimental evaluation of differential privacy. The notion of differential privacy (Dwork & Roth (2014)) appears naturally in many applications when it comes to estimating of the possibility of privacy violation. In particular, it can be applied to language models trained on private user data.

Loosely speaking, if we have a mechanism that takes some input data and produces some output then differential privacy measures how a single input unit influences the total output. In order to achieve differential privacy, some randomness must be introduced into the mechanism.

**Definition 1.** *A randomized mechanism $\mathcal{M}$ with domain $\mathcal{D}$ and range $\mathcal{S}$ satisfies $(\varepsilon, \delta)$-differential privacy if for any two inputs $d, d' \in \mathcal{D}$ that are adjacent (i.e. differ in one record) and for any subset of outputs $S \subseteq \mathcal{S}$ it holds that:*

$$P(\mathcal{M}(d) \in S) \leq e^{\varepsilon} P(\mathcal{M}(d') \in S) + \delta$$

In our case $\mathcal{D}$ is the set of all subsets of users and a randomized mechanism $\mathcal{M}(d)$ is a mechanism that generates texts according to a certain language model trained on $d \in \mathcal{D}$. Note that for any $d$ we need to have

$$\sum_{s \in \mathcal{S}} P(\mathcal{M}(d) = s) = 1$$

Thus it is necessary for $\mathcal{S}$ to be the set of all possible texts of some fixed length rather than the set of all texts of an arbitrary length. In our analysis we will consider only the space of texts containing 10 words. This is reasonable because it is close to the average length of a sentence in our user data corpus and it seems that if user's privacy is violated then 10 consequent words are already enough for an adversary to retrieve important information.

Let us fix two adjacent sets of users $d$ and $d'$, train models $\theta$ and $\theta'$ on them and introduce random variable $c(s)$. It is defined by the expression

$$c(s) = \frac{P(s|\theta)}{P(s|\theta')} \tag{6}$$

for any $s \in \mathcal{S}$. Since a language model $\Theta$ assigns some positive probability to any sequence of words, $c(s)$ is defined correctly for all $s \in \mathcal{S}$.

Parameter $\delta$ in the Definition 1 stands for the probability that two probabilities $P(s|\theta)$ and $P(s|\theta')$ differ much. This fact is formalized by the following proposition:

**Proposition 1.** *If $P(s \in \mathcal{S} : c(s) > e^{\varepsilon} \, |\theta) \leq \delta$ then $P(S|\theta) \leq e^{\varepsilon} P(S|\theta') + \delta$ for any $S \subseteq \mathcal{S}$*

*Proof.* Let $B = \{s \in \mathcal{S} : c(s) > e^{\varepsilon}\}$. Then for any $S \subseteq \mathcal{S}$

$$P(S|\theta) = P(S \cap B|\theta) + P(S \cap \overline{B}|\theta) \leq P(B|\theta) + e^{\varepsilon} P(S \cap \overline{B}|\theta') \leq \delta + e^{\varepsilon} P(S|\theta')$$

$\square$

The proposition implies that it is sufficient to estimate the tail of the distribution of $c(s)$ under measure $\mathbb{P}(\cdot|\theta)$. Furthermore, Figure 3 suggests that the tail of the empirical distribution function of the observed variable $c(s)$ has the Pareto distribution. This seems natural as far as words in human language follow Zipf's law which is a discrete analogue of the Pareto distribution.

To make a confident estimation of differential privacy parameters, we consider 20 different pairs of adjacent sets of users $d$ and $d'$. For each one, we consider a composite null hypothesis that the tail of

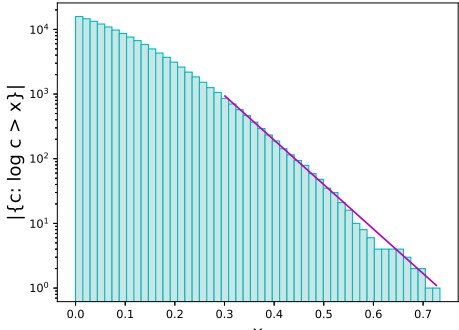 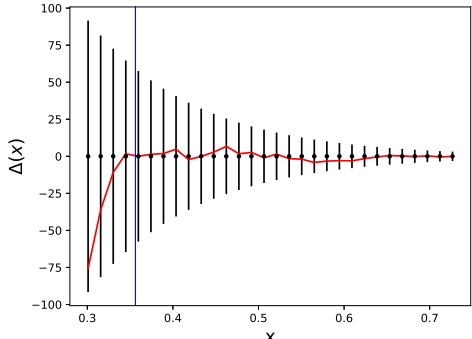

Figure 3: *Left:* Empirical histogram of random samples of $c(s)$. Magenta line represents theoretical distribution of the Pareto law with parameters that are estimated on these samples. *Right:* Difference between two distributions on the left plot expressed in number of samples $\Delta(x)$. The parameters of the Pareto law were estimated on the samples that lie in the region $\{\log c(s) > 0.35\}$ (blue line). Black lines represent standard errors. The left plot is built in logarithmic Y-axis while the right one is built in linear Y-axis.

the random variable $c(s)$ defined in (6) has the Pareto distribution with the shape parameter equal to its Hill's estimator (M. Hill (1975)). Then we apply the Lilliefors test and accept the null hypothesis at a significance level of $5\%$. Quantiles of the Pareto distribution can be written down explicitly thus giving the following formula for estimation of parameters $\varepsilon$ and $\delta$:

$$\varepsilon = \frac{1}{\alpha} \log \frac{C}{\delta}, \tag{7}$$

where $\alpha$ and $C$ are parameters of Pareto distribution defined in statistical tests (see Appendix).

Finally, for a given $\delta$ we take the largest value of $\varepsilon$ amongst all the experiments.

## 3.2 EXPERIMENTAL EVALUATION

The critical value for the Lilliefors test at $5\%$ significance level is $1.08$. In 19 cases out of 20 the Lilliefors test fails to reject the null hypothesis. This conclusion, together with sample visual representation in Figure 3, allows us to state that the random variable $c(s)$ indeed has tails that decrease like the Pareto distribution tails with quite a big shape parameter. Exact values of KS statistics and Hill's estimators of this parameter for different pairs of users are provided in the Table 5.

Table 6 shows the results for different values of $\delta$ calculated by formula (7). In this table the value of $\varepsilon$ is the largest value of this parameter in all 20 experiments. The total number of users is $3 \cdot 10^3$ so it is reasonable to put $\delta = 10^{-4}$. For this choice of $\delta$ parameter $\varepsilon$ equals to $0.67$. It means that our algorithm offers reasonable privacy guarantees (see (Papernot et al., 2017)). Additionally we provide values of $\varepsilon$ for smaller values of $\delta$.

The results shown in Table 6 demonstrate that our scheme provides a very good level of privacy protection. However, it is necessary to say that we only aim to produce an empirical estimation of differential privacy which inevitably holds with some high probability but not almost surely (this fact makes our approach close to the so-called *random differential privacy* introduced in Hall et al. (2011)). In many machine learning algorithms, the outcome is initially deterministic and some well-known distribution is used to generate noise in order to make the algorithm differentially private (e.g. Papernot et al. (2017)). In our mechanism the source of randomness lies inside the neural network and the output distributions can't be written explicitly. This is the reason why we are able to provide only empirical estimations of differential privacy parameters.

Table 5: Results of the Lilliefors test

| Experiment | 1 | 2 | 3 | 4 | 5 | 6 | 7 | 8 | 9 | 10 |
|---|---|---|---|---|---|---|---|---|---|---|
| $\widehat{\alpha}$ | 15.8 | 20.9 | 15.1 | 16.6 | 16.5 | 17.6 | 14.9 | 19.2 | 15.6 | 15.2 |
| $\widehat{C}$ | 3.25 | 5.64 | 2.02 | 2.48 | 2.70 | 4.19 | 1.47 | 3.31 | 1.65 | 1.83 |
| KS statistic | 0.49 | 0.91 | 0.48 | 0.62 | 0.83 | 0.59 | **1.39** | 0.41 | 0.93 | 0.51 |
| Experiment | 11 | 12 | 13 | 14 | 15 | 16 | 17 | 18 | 19 | 20 |
| $\widehat{\alpha}$ | 16.5 | 14.4 | 19.5 | 18.2 | 16.2 | 17.2 | 17.3 | 14.8 | 17.1 | 20.5 |
| $\widehat{C}$ | 3.00 | 1.53 | 3.67 | 2.20 | 3.42 | 2.66 | 1.68 | 2.18 | 2.87 | 4.60 |
| KS statistic | 0.76 | 0.89 | 0.66 | 0.94 | 0.67 | 0.85 | 0.73 | 0.97 | 0.65 | 0.94 |

Table 6: Differential privacy results

| $\delta$ | $10^{-4}$ | $10^{-5}$ | $10^{-6}$ |
|---|---|---|---|
| $\varepsilon$ | 0.67 | 0.83 | 0.99 |

## 4 CONCLUSION

We have presented our results in distributed fine-tuning of neural language models. We paid special attention to preventing a catastrophic forgetting of the general language after a model fine-tuning on the user devices. Our experiments showed that the performance of an initial model of the general English on user data can be improved significantly almost without a performance degradation on the standard English training data. We found that a combination of on-device training with random rehearsal and server-side model averaging provides the best performance for such distributed fine-tuning. Users' models were trained for the whole epoch that reduced communication costs while at the same time being quite fast – it took less than 3 minutes with a realistic assessment of volume of the available user data. Finally, we provided an experimental evaluation of differential privacy of our method and showed that the method has a reasonable level of differential privacy compared to other solutions. We still have to note that we provided an empirical estimation of differential privacy which holds with some high probability but not almost surely.

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

## A    EXPERIMENTAL EVALUATION OF DIFFERENTIAL PRIVACY FOR TEXTS

One can usually identify that samples come from a power-law distribution by looking at its tail distribution function $\overline{F}(x) = 1 - F(x)$ where $F(x)$ is a cumulative distribution function (e.g. Newman (2005) describes this method). If $\overline{F}(x) = C/x^\alpha$ then $\log \overline{F}(x) = \log C - \alpha \log x$, i.e. the plot should be linear on logarithmic axes.

Figure 3 shows the empirical tail distribution function of the observed variable $c(s)$. We generated $n = 3 \cdot 10^4$ samples (10-word sequences) with the model with parameters $\theta$ that relates to a certain user to get observations of $c(s)$. It can be seen that the tail of $c(s)$ is linear on logarithmic axes like the tail of the Pareto distribution in the region $\{\log c(s) > 0.35\}$.

So we suppose that $\overline{F}(x) = C/x^\alpha$ for big values of $x$. More precisely, we suppose that the distribution function of $c(s)$ for $x > x_0$ can be represented by the following formula:

$$F(x) = 1 - \overline{F}(x_0) \cdot \left(\frac{x_0}{x}\right)^\alpha \tag{8}$$

for some $x_0$. Parameter $\alpha$ plays the most important role in the further analysis of differential privacy. A common way to estimate it is to use Hill's estimator:

$$\widehat{\alpha} = \frac{k}{\sum_{i=1}^{k} \log \frac{c_n^{(i)}}{c_n^{(k)}}} \tag{9}$$

where $c_n^{(i)}$ are the order statistics $c_n^{(1)} \geq c_n^{(2)} \geq ... \geq c_n^{(k)} \geq ... \geq c_n^{(n)}$ and $n$ is the number of samples. This estimator is described in M. Hill (1975). It is a maximum likelihood estimator and it converges in probability to $\alpha$ when $n \to \infty$, $k = k(n) \to \infty$ and $k(n)/n \to 0$. Note that the estimator depends only on outliers of $c(s)$. This is a very helpful property because it allows us to use it even when we need to estimate only the tail distribution $\overline{F}(x)$ for large values of $x$ rather than the whole distribution. In the experiments we take $k(n) = 2 \lfloor \sqrt{n} \rfloor$. We put $x_0 = c_n^{(k)}$. For different pairs of adjacent sets of users $d$ and $d'$ values of $\widehat{\alpha}$ vary from 14.4 to 20.9. Values of $x_0$ vary from 1.33 to 1.43, so $\log x_0$ lies in the interval $[0.28; 0.36]$ in our experiments.

Then we tested the null hypothesis that the cumulative distribution function $F(x)$ of the random variable $c(s)$ is of the Pareto law with the shape parameter $\widehat{\alpha}$ for all $x > x_0$. The Kolmogorov-Smirnov (KS) test is often used for this purpose (Koning & Peng (2008) illustrates this approach). Since we tested a composite hypothesis, we needed to use modification of the KS test that is called the Lilliefors test. In the same way as in Koning & Peng (2008) we introduced new random variables $r_i = \log c_n^{(i)}/c_n^{(k)}$ for $i = 1, .., k$. Since $c_n^{(i)}$ are order statistics, we have $c_n^{(i)}/c_n^{(k)} \geq 1$ for $i = 1, .., k$ and it can be shown that these variables are jointly equal in distribution to ordered samples from Pareto law with the shape parameter $\alpha$ and the scale parameter 1. So, under the null hypothesis $\{r_i\}_{1,...,k}$ are exponential with the parameter $\alpha$ and we can apply the Lilliefors test to check whether these samples really come from an exponential distribution with an unknown mean estimated by $\overline{r} = 1/\widehat{\alpha}$.

The method that we use (the Lilliefors test for exponential distributions) is described in Gibbons & Chakraborti (2010). Essentially, we calculate a KS statistic for the exponential distribution with a mean that's equal to 1 and an empirical distribution function $F_k(r)$ of the values $\{r_i/\overline{r}\}_{1,..,k}$:

$$\sqrt{k} \sup_{r \geq 1} |F_k(r) - (1 - e^{-r})| \tag{10}$$

This statistic doesn't converge to the Kolmogorov distribution as shown in W. Lilliefors (1969). It converges to the distribution with smaller critical values at the same significance levels because we overfit on the sample data when the estimator $\bar{r}$ is plugged in. We chose a $5\%$ significance level and critical value for it is $1.08$. In 19 cases out of 20 the Lilliefors test failed to reject the null hypothesis at a $5\%$ significance level. Table 5 provides exact values obtained during the application of the statistical test. Relying on these values along with data visualization in 3 we can state that random variable $c(s)$ has tails that decrease like the Pareto distribution tails.

The hypothesis that we accepted suggests that the cumulative distribution function of $c(s)$ is given by the formula (8). It means that the tail distribution function for all $x > x_0$ is given by

$$\overline{F}(x) = \frac{\overline{F}(x_0)x_0^\alpha}{x^\alpha} = \frac{C}{x^\alpha} \tag{11}$$

We chose $x_0 = c_n^{(k)}$, so $\overline{F}(x_0)$ is just the ratio $k/n$. Thus, $C$ can be estimated by

$$\widehat{C} = \frac{k}{n} \cdot (c_n^{(k)})^{\widehat{\alpha}} \tag{12}$$

Values of $\widehat{C}$ are given in the Table 5. Finally, from formula (11) and proposition 1 it is easy to derive that $(\varepsilon, \delta)$-differential privacy is provided by the values $\varepsilon$, $\delta$ that satisfy

$$\varepsilon = \frac{1}{\alpha} \log \frac{C}{\delta} \tag{13}$$

