# OpenReview forum: "Distributed Fine-tuning of Language Models on Private Data"
_ICLR.cc/2018/Conference — Accept (Poster)_

### Official Review · AnonReviewer1 · 2017-11-27
**More explanations on the assumption are required**

**Rating:** 5
**Confidence:** 4

**Review:**

This paper deals with improving language models on mobile equipments
based on small portion of text that the user has ever input. For this
purpose, authors employed a linearly interpolated objectives between user
specific text and general English, and investigated which method (learning
without forgetting and random reheasal) and which interepolation works better.
Moreover, authors also look into privacy analysis to guarantee some level of
differential privacy is preserved.

Basically the motivation and method is good, the drawback of this paper is
its narrow scope and lack of necessary explanations. Reading the paper,
many questions arise in mind:

- The paper implicitly assumes that the statistics from all the users must
  be collected to improve "general English". Why is this necessary? Why not
  just using better enough basic English and the text of the target user?

- To achieve the goal above, huge data (not the "portion of the general English") should be communicated over the network. Is this really worth doing? If only
  "the portion of" general English must be communicated, why is it validated?

- For measuring performance, authors employ keystroke saving rate. For the
  purpose of mobile input, this is ok: but the use of language models will
  cover much different situation where keystrokes are not necessarily
  available, such as speech recognition or machine translation. Since this
  paper is concerned with a general methodology of language modeling,
  perplexity improvement (or other criteria generally applicable) is also
  important.

- There are huge number of previous work on context dependent language models,
  let alone a mixture of general English and specific models. Are there any
  comparison with these previous efforts?

Finally, this research only relates to ICLR in that the language model employed
is LSTM: in other aspects, it easily and better fit to ordinary NLP conferences, such as EMNLP, NAACL or so. I would like to advise the authors to submit
this work to such conferences where it will be reviewed by more NLP experts.

Minor:
- t of $G_t$ in page 2 is not defined so far.
- What is "gr" in Section 2.2?

---

> ### Public Comment · (anonymous) · 2017-12-05
> **Answer to the review and some clarifications**
>
> Thank you for your review!
> I would like to make some clarifications and remarks.
>
> You write:
> "- The paper implicitly assumes that the statistics from all the users must be collected to improve "general English". Why is this necessary? Why not just using better enough basic English and the text of the target user?"
>
> There is strong evidence that  the language of SMS and/or private messaging is sufficiently different from what we can collect in publicly available resources. Since language changes constantly we need to update the models and we cannot just make a single fine-tuning of basic LM on device. On the other hand the data from a single user is not sufficient for model update so we need data from many different users. The problem is that we cannot (or at least don't want to) collect user data.  We've proposed a method of continuous update of language models without need to collect private data.
>
> "- To achieve the goal above, huge data (not the "portion of the general English") should be communicated over the network. Is this really worth doing? If only the portion of" general English must be communicated, why is it validated?"
>
> As we mention in the paper the volume of the user generated data is small. Actually users generate approx. 600 bytes/day. In our experiments we proceeded from the assumption that fine-tuning starts as soon as 10 Kb of text data is accumulated on device. Our experiments showed that random rehearsal with volume of the rehearsal data should be to the volume of the fine-tuning data. So it is 10 Kb. This number is very small compared to the volume of model weights which are communicated in Federated Learning algorithms. We also discussed the communication efficiency in the answers to other reviewers (see above).
>
> "- For measuring performance, authors employ keystroke saving rate. For the purpose of mobile input, this is ok: but the use of language models will cover much different situation where keystrokes are not necessarily available, such as speech recognition or machine translation. Since this paper is concerned with a general methodology of language modeling, perplexity improvement (or other criteria generally applicable) is also important."
>
> Basically, we agree. And perplexity is reported in all our experiments. We just wanted to emphasize that target metrics should also be evaluated in language modeling like it is done in speech recognition (ASR) or machine translation (BLEU). Also, in (McMahan et al. 2017, https://openreview.net/pdf?id=B1EPYJ-C-) word prediction accuracy is evaluated which is relative to KSS.
>
> "- There are huge number of previous work on context dependent language models, let alone a mixture of general English and specific models. Are there any comparison with these previous efforts?"
>
>
> The term context is a bit vague. E.g. in (Mikolov, Zweig 2014, http://citeseerx.ist.psu.edu/viewdoc/download?doi=10.1.1.258.5120&rep=rep1&type=pdf) term "context" refers to the longer left context which is unavailable to standard RNN. Anyway longer left contexts are reasonably well catched by LSTM. If "context" refers to the running environment (e.g. application context) it is not the exact scope of our work. The standard approach to model adaptaion for the user is either model fine-tuning or interpolation with simpler language model (e.g. Knesser-Ney smoothed n-gram). We tried approaches similar to the proposed in (Ma et al. 2017, https://static.googleusercontent.com/media/research.google.com/ru//pubs/archive/46439.pdf) but they performed significantly worse.
>
> I also would like to draw your attention to the privacy analysys part of the paper which we included in the list of our contributions. We consider our contribution significant at least for the following reason. To our knowledge deep neural networks have never been checked for differential privacy coming from the randomness of the training algorithm (combination of SGD, dropout regularization and model averaging in our case). Existing approaches (e.g. Papernot et al. 2017, https://arxiv.org/pdf/1610.05755.pdf) suggest adding random noise at different stages of training leading to the tradeoff between accuracy and privacy. At the same time our experiments show that the differential privacy can be guaranteed even without special treatment of the neural networks at least in some situations.

---

> ### Author Response · Authors · 2017-12-20
> **Answer to the review and some clarifications**
>
> Thank you for your review!
> I would like to make some clarifications and remarks.
>
> You write:
> "- The paper implicitly assumes that the statistics from all the users must be collected to improve "general English". Why is this necessary? Why not just using better enough basic English and the text of the target user?"
>
> There is strong evidence that  the language of SMS and/or private messaging is sufficiently different from what we can collect in publicly available resources. Since language changes constantly we need to update the models and we cannot just make a single fine-tuning of basic LM on device. On the other hand the data from a single user is not sufficient for model update so we need data from many different users. The problem is that we cannot (or at least don't want to) collect user data.  We've proposed a method of continuous update of language models without need to collect private data.
>
> "- To achieve the goal above, huge data (not the "portion of the general English") should be communicated over the network. Is this really worth doing? If only the portion of" general English must be communicated, why is it validated?"
>
> As we mention in the paper the volume of the user generated data is small. Actually users generate approx. 600 bytes/day. In our experiments we proceeded from the assumption that fine-tuning starts as soon as 10 Kb of text data is accumulated on device. Our experiments showed that random rehearsal with volume of the rehearsal data should be to the volume of the fine-tuning data. So it is 10 Kb. This number is very small compared to the volume of model weights which are communicated in Federated Learning algorithms. We also discussed the communication efficiency in the answers to other reviewers (see above).
>
> "- For measuring performance, authors employ keystroke saving rate. For the purpose of mobile input, this is ok: but the use of language models will cover much different situation where keystrokes are not necessarily available, such as speech recognition or machine translation. Since this paper is concerned with a general methodology of language modeling, perplexity improvement (or other criteria generally applicable) is also important."
>
> Basically, we agree. And perplexity is reported in all our experiments. We just wanted to emphasize that target metrics should also be evaluated in language modeling like it is done in speech recognition (ASR) or machine translation (BLEU). Also, in (McMahan et al. 2017, https://openreview.net/pdf?id=B1EPYJ-C-) word prediction accuracy is evaluated which is relative to KSS.
>
> "- There are huge number of previous work on context dependent language models, let alone a mixture of general English and specific models. Are there any comparison with these previous efforts?"
>
>
> The term context is a bit vague. E.g. in (Mikolov, Zweig 2014, http://citeseerx.ist.psu.edu/viewdoc/download?doi=10.1.1.258.5120&;rep=rep1&type=pdf) term "context" refers to the longer left context which is unavailable to standard RNN. Anyway longer left contexts are reasonably well catched by LSTM. If "context" refers to the running environment (e.g. application context) it is not the exact scope of our work. The standard approach to model adaptaion for the user is either model fine-tuning or interpolation with simpler language model (e.g. Knesser-Ney smoothed n-gram). We tried approaches similar to the proposed in (Ma et al. 2017, https://static.googleusercontent.com/media/research.google.com/ru//pubs/archive/46439.pdf) but they performed significantly worse.
>
> I also would like to draw your attention to the privacy analysys part of the paper which we included in the list of our contributions. We consider our contribution significant at least for the following reason. To our knowledge deep neural networks have never been checked for differential privacy coming from the randomness of the training algorithm (combination of SGD, dropout regularization and model averaging in our case). Existing approaches (e.g. Papernot et al. 2017, https://arxiv.org/pdf/1610.05755.pdf) suggest adding random noise at different stages of training leading to the tradeoff between accuracy and privacy. At the same time our experiments show that the differential privacy can be guaranteed even without special treatment of the neural networks at least in some situations.

---

### Official Review · AnonReviewer2 · 2017-11-28
**The relevance of this paper to ICLR is low and the details of the experiments are missing.**

**Rating:** 4
**Confidence:** 3

**Review:**

my main concern is the relevance of this paper to ICLR.
This paper is much related not to representation learning but to user-interface.
The paper is NOT well organized and so the technical novelty of the method is unclear.
For example, the existing method and proposed method seems to be mixed in Section 2.
You should clearly divide the existing study and your work.
The experimental setting is also unclear.
KSS seems to need the user study.
But I do not catch the details of the user study, e.g., the number of users.

---

> ### Public Comment · (anonymous) · 2017-12-04
> **Answer to the review and some clarifications**
>
> Thank you for your review!
> Below I will try to answer to your remarks.
>
> 1) You write
> "my main concern is the relevance of this paper to ICLR"
> "This paper is much related not to representation learning but to user-interface"
> We think that our paper is relevant for ICLR because the papers on the same or adjacent topics were or will be presented:
> 1. https://openreview.net/forum?id=SkhQHMW0W (ICLR 2018, federated learning)
> 2. https://arxiv.org/pdf/1610.05755.pdf (ICLR 2017, differential privacy)
> 3. https://openreview.net/forum?id=BJ0hF1Z0b&noteId=Bkg5_kcxG (ICLR 2018, differentially private RNN)
> We admit that our field of study is not as broad because we work only with language models, but the approach proposed in the paper may be used to different types of data and ML tasks. Our method gives good privacy guarantees and provides low communication cost (compared to previous results).
>
> Let me cite my answer to the previous reviewer.
>
> "In the works on Federated Learning issued so far each node is considered only as a client in distributed learning system for gradient calculation. In our approach we guarantee that the model sent to the aggregation server is at the same time the actual model used for typing and gives the best performance for the end user. It is guaranteed by the forgetting prevention mechanism. It has at least following advantages: 1) No need  for synchronization after every iteration as in standard Federated learning scheme. Standard Federated learning uses no more than 20 iterations on each device for reduction of the communication cost (McMahan et al. 2016, https://arxiv.org/abs/1602.05629) while we send our models only once in an epoch thus significantly reducing the communication cost. ; 2) Simpler synchronization scheme on the server; 3) Faster convergence; 4) Only 1 model is stored on the disk. We think that these results may be interesting to many ML practitioners."
>
> 2) "KSS seems to need the user study". - KSS is measured according to the formula (5) given in the paper. The process of testing automatization seems obvious.  No user testing is needed. (comp. WER calculation for ASR). In any case perplexity results are also given for all experiments.
>
> 3) As far as you didn't discuss the privacy analysis part which was included into the contributions list I'll cite my previous comment again:
>
> "We consider our contribution significant at least for the following reason. To our knowledge deep neural networks have never been checked for differential privacy coming from the randomness of the training algorithm (combination of SGD, dropout regularization and model averaging in our case). Existing approaches (e.g. Papernot et al. 2017, https://arxiv.org/pdf/1610.05755.pdf) suggest adding random noise at different stages of training leading to the tradeoff between accuracy and privacy. At the same time our experiments show that the differential privacy can be guaranteed even without special treatment of the neural networks at least in some situations."

---

> ### Author Response · Authors · 2017-12-20
> **Answer to the review and some clarifications**
>
> Thank you for your review!
> Below I will try to answer to your remarks.
>
> 1) You write
> "my main concern is the relevance of this paper to ICLR"
> "This paper is much related not to representation learning but to user-interface"
> We think that our paper is relevant for ICLR because the papers on the same or adjacent topics were or will be presented:
> 1. https://openreview.net/forum?id=SkhQHMW0W (ICLR 2018, federated learning)
> 2. https://arxiv.org/pdf/1610.05755.pdf (ICLR 2017, differential privacy)
> 3. https://openreview.net/forum?id=BJ0hF1Z0b&;noteId=Bkg5_kcxG (ICLR 2018, differentially private RNN)
> We admit that our field of study is not as broad because we work only with language models, but the approach proposed in the paper may be used to different types of data and ML tasks. Our method gives good privacy guarantees and provides low communication cost (compared to previous results).
>
> Let me cite my answer to the previous reviewer.
>
> "In the works on Federated Learning issued so far each node is considered only as a client in distributed learning system for gradient calculation. In our approach we guarantee that the model sent to the aggregation server is at the same time the actual model used for typing and gives the best performance for the end user. It is guaranteed by the forgetting prevention mechanism. It has at least following advantages: 1) No need  for synchronization after every iteration as in standard Federated learning scheme. Standard Federated learning uses no more than 20 iterations on each device for reduction of the communication cost (McMahan et al. 2016, https://arxiv.org/abs/1602.05629) while we send our models only once in an epoch thus significantly reducing the communication cost. ; 2) Simpler synchronization scheme on the server; 3) Faster convergence; 4) Only 1 model is stored on the disk. We think that these results may be interesting to many ML practitioners."
>
> 2) "KSS seems to need the user study". - KSS is measured according to the formula (5) given in the paper. The process of testing automatization seems obvious.  No user testing is needed. (comp. WER calculation for ASR). In any case perplexity results are also given for all experiments.
>
> 3) As far as you didn't discuss the privacy analysis part which was included into the contributions list I'll cite my previous comment again:
>
> "We consider our contribution significant at least for the following reason. To our knowledge deep neural networks have never been checked for differential privacy coming from the randomness of the training algorithm (combination of SGD, dropout regularization and model averaging in our case). Existing approaches (e.g. Papernot et al. 2017, https://arxiv.org/pdf/1610.05755.pdf) suggest adding random noise at different stages of training leading to the tradeoff between accuracy and privacy. At the same time our experiments show that the differential privacy can be guaranteed even without special treatment of the neural networks at least in some situations."

---

### Official Review · AnonReviewer3 · 2017-11-30
**Very relevant for the application in mind. Too simple for publication.**

**Rating:** 4
**Confidence:** 4

**Review:**

This paper discusses the application of word prediction for software keyboards. The goal is to customize the predictions for each user to account for member specific information while adhering to the strict compute constraints and privacy requirements.

The authors propose a simple method of mixing the global model with user specific data. Collecting the user specific models and averaging them to form the next global model.

The proposal is practical. However, I am not convinced that this is novel enough for publication at ICLR.

One major question. The authors assume that the global model will depict general english. However, it is not necessary that the population of users will adhere to general English and hence the averaged model at the next time step t+1 might be significantly different from general English. It is not clear to me as how this mechanism guarantees that it will not over-fit or that there will be no catastrophic forgetting.

---

> ### Public Comment · (anonymous) · 2017-12-04
> **Answer to the review and some clarifications**
>
> Thank you for your review!
> I would like to make some clarifications and remarks.
>
> 1) In the review you write
> "One major question. The authors assume that the global model will depict general english. However, it is not necessary that the population of users will adhere to general English and hence the averaged model at the next time step t+1 might be significantly different from general English. It is not clear to me as how this mechanism guarantees that it will not over-fit or that there will be no catastrophic forgetting."
>
> In our problem statement we consider general English to be the common language i.e. the commonly used language with statistically insignificant portion of user-specific expressions. It is NOT necessarily the language induced by our in-house corpus. As far as we have a model averaged on many users on the server we can treat this model as general language model. So at each stage T the server-side model represents general language. When the model is sent to the device it is updated according to the user data so there is a risk of catastrophic forgetting. We prevent it by using (eventually) random rehearsal on device (only!).
>
> 2) I also would like to draw your attention to the original and practically relevant problem statement. In the works on Federated Learning issued so far each node is considered only as a client in distributed learning system for gradient calculation. In our approach we guarantee that the model sent to the aggregation server is at the same time the actual model used for typing and gives the best performance for the end user. It is guaranteed by the forgetting prevention mechanism. It has at least following advantages: 1) No need  for synchronization after every iteration as in standard Federated learning scheme. Standard Federated learning uses no more than 20 iterations on each device for reduction of the communication cost (McMahan et al. 2016, https://arxiv.org/abs/1602.05629) while we send our models only once in an epoch thus significantly reducing the communication cost. ; 2) Simpler synchronization scheme on the server; 3) Faster convergence; 4) Only 1 model is stored on the disk. We think that these results may be interesting to many ML practitioners.
>
> 3) You didn't discuss the privacy analysis part of the paper which we included in the list of our contributions. We consider our contribution significant at least for the following reason. To our knowledge deep neural networks have never been checked for differential privacy coming from the randomness of the training algorithm (combination of SGD, dropout regularization and model averaging in our case). Existing approaches (e.g. Papernot et al. 2017, https://arxiv.org/pdf/1610.05755.pdf) suggest adding random noise at different stages of training leading to the tradeoff between accuracy and privacy. At the same time our experiments show that the differential privacy can be guaranteed even without special treatment of the neural networks at least in some situations.

---

> ### Author Response · Authors · 2017-12-20
> **Answer to the review and some clarifications**
>
> Thank you for your review!
> I would like to make some clarifications and remarks.
>
> 1) In the review you write
> "One major question. The authors assume that the global model will depict general english. However, it is not necessary that the population of users will adhere to general English and hence the averaged model at the next time step t+1 might be significantly different from general English. It is not clear to me as how this mechanism guarantees that it will not over-fit or that there will be no catastrophic forgetting."
>
> In our problem statement we consider general English to be the common language i.e. the commonly used language with statistically insignificant portion of user-specific expressions. It is NOT necessarily the language induced by our in-house corpus. As far as we have a model averaged on many users on the server we can treat this model as general language model. So at each stage T the server-side model represents general language. When the model is sent to the device it is updated according to the user data so there is a risk of catastrophic forgetting. We prevent it by using (eventually) random rehearsal on device (only!).
>
> 2) I also would like to draw your attention to the original and practically relevant problem statement. In the works on Federated Learning issued so far each node is considered only as a client in distributed learning system for gradient calculation. In our approach we guarantee that the model sent to the aggregation server is at the same time the actual model used for typing and gives the best performance for the end user. It is guaranteed by the forgetting prevention mechanism. It has at least following advantages: 1) No need  for synchronization after every iteration as in standard Federated learning scheme. Standard Federated learning uses no more than 20 iterations on each device for reduction of the communication cost (McMahan et al. 2016, https://arxiv.org/abs/1602.05629) while we send our models only once in an epoch thus significantly reducing the communication cost. ; 2) Simpler synchronization scheme on the server; 3) Faster convergence; 4) Only 1 model is stored on the disk. We think that these results may be interesting to many ML practitioners.
>
> 3) You didn't discuss the privacy analysis part of the paper which we included in the list of our contributions. We consider our contribution significant at least for the following reason. To our knowledge deep neural networks have never been checked for differential privacy coming from the randomness of the training algorithm (combination of SGD, dropout regularization and model averaging in our case). Existing approaches (e.g. Papernot et al. 2017, https://arxiv.org/pdf/1610.05755.pdf) suggest adding random noise at different stages of training leading to the tradeoff between accuracy and privacy. At the same time our experiments show that the differential privacy can be guaranteed even without special treatment of the neural networks at least in some situations.

---

### Decision · Program_Chairs · 2018-01-29
**ICLR 2018 Conference Acceptance Decision**

**Decision:**

Accept (Poster)

**Comment:**

The committee feels that this paper presents a simple, yet effective way to adapt language models from various users in a sufficiently privacy preserving way.  Empirical results are quite strong.  Reviewer 3 says that the novelty of the paper is not great, but does not provide any references to prior work that are similar to this paper.  The meta-reviewer finds the responses to Reviewer 3 sufficient to address the concerns.

Similarly, Reviewer 2 says that the paper may not be relevant to ICLR, but the committee feels its content does belong to the conference since the topic is extremely relevant to modern language processing techniques.  In fact, the authors provide several references that show that this paper is similar in content to those submissions.

Reviewer 1's concerns are also not sufficiently strong to warrant rejection.  The responses to each criticism suffices and the meta-reviewer thinks that this paper will add value to the conference.